# Dynamic Analysis of Sphere-Like Iron Particles Based Magnetorheological Damper for Waveform-Generating Test System

**DOI:** 10.3390/ijms21031149

**Published:** 2020-02-09

**Authors:** Jong-Seok Oh, Chang Won Shul, Tae Hyeong Kim, Tae-Hoon Lee, Sung-Wan Son, Seung-Bok Choi

**Affiliations:** 1Division of Mechanical and Automotive Engineering, Kongju National University, Chungnam 31080, Korea; 2Agency for Defense Development, Daejeon 305-600, Korea; shulcw@add.re.kr (C.W.S.); kimth@add.re.kr (T.H.K.); 3Department of Mechanical Engineering, Smart Structures and Systems Laboratory, Inha University, Incheon 22212, Korea; kakhangl@naver.com; 4RMS Technology, Pungse Industrial Complex A17-6, Boseong-ri, Chungcheongnam-do 31217, Korea; swson@rmstech.co.kr

**Keywords:** Magnetorheological (MR) Fluid, MR Damper, Double Waveform-generating test System, Shock-Wave Profile

## Abstract

In this study, a new double pulse waveform-generating test system with an integrated magnetorheological (MR) damper is proposed. Since the total shear stress of MR fluid can be varied according to the shape of particles, sphere-like iron particles-based MR fluid is filled into the MR damper. The test system consists of a velocity generator, three masses (impact, test, and dummy), a spring, and an MR damper. To tune the double pulse waveform profile, a damping force model is constructed to determine the fundamental parameters of the simulator. Then, the first and second shock waveform profiles are analyzed to solve the governing equation of motions representing the damping force and velocity. The mathematical model of the MR damper is formulated and applied to a simulator with a graphical user interface programmed using MATLAB. The effectiveness of the proposed simulator-featuring controllable MR damper is demonstrated by comparing the simulation and experimental results.

## 1. Introduction

The first discovery of a magnetorheological (MR) fluid was made by Jacob Rainbow in the late 1940s [1,2]. Typically, MR fluids are composed of paramagnetic or soft ferromagnetic particles (~0.03–10 μm) dispersed in a carrier fluid. The MR particles are made of pure iron, carbonyl iron, or cobalt powder, and the carrier fluid is a non-magnetic, organic, or aqueous liquid, usually a silicone or mineral oil [3]. Without a magnetic field, the MR particles are randomly distributed in the carrier fluid. However, in magnetic field, they undergo rheological phase changes according to the magnitude of the field as shown in Figure 1a. The MR particles attain a dipole moment aligned with the external magnetic field and form chains. This chain formation causes an additional yield stress in the fluid, which can be rapidly and continuously controlled by magnetic input. As shown in Figure 1b, it is generally known that maximum yield stress of MR fluid is approximately 60–80 kPa. The power requirement for magnetic inputs is up to 50 Watts. In addition, it has been reported that the response time of commercial MR fluid is below 20 ms. As a result, MR fluid-based mechanical systems have salient advantages such as continuously adjustable dynamic range and fast response. However, the performance of MR fluid can be varied by the particle shape of MR fluid. As shown in Figure 1c, the shapes of particles are mostly sphere shape type and plate-lie shape type. Some of the research has analyzed the effect of particle shape of MR fluid [4,5,6,7]. From the previous work, it is revealed that sphere-like iron particles-based MR fluid is proper for MR damper, generating high damping force [8]. So, sphere-like iron particles-based MR fluid (RMS Corp., T5-500) is used in this work.

Owing to the properties of MR fluids, MR dampers have inherent advantages such as fast response time and continuously controllable damping force. MR dampers are used in application devices, including semi-active controllable devices, such as shock absorbers for vehicle suspension systems [9,10,11,12,13,14], landing gears [15,16], and seismic dampers [17,18]. At present, shock wave profile generators based on MR dampers are actively researched. Many studies have been undertaken to evaluate the durability of on-board ship equipment using random shock waves generated from underwater explosions [19]. W. Zhaodong et al. [20] suggested a dual-wave shock-testing machine to simulate a non-contact underwater explosion and test the survivability of shipboard equipment to shocks. G. Wang et al. [21] investigated a vertical heavyweight shock test machine. Dynamic modeling and a mechanism analysis verified that the proposed test machine can produce shock acceleration wave profiles consistent with the MIL-S-901D criteria (via computer simulation). C. W. Shul et al. [22,23,24] carried out a dynamic design analysis for heavy dual shock generation systems. However, until now, a complete determination of the necessary parameter values of testing machines has not been possible because of the nonlinearity of large-scale MR dampers. 

In this study, the results of a performance evaluation for a single unit large-scale MR damper were used to establish a mathematical model that can express the nonlinearity and to determine the set values of each element needed to realize the desired waveform. To obtain the desired shock wave magnitude and duration for each waveform in the double pulse waveform, the MR damper was employed to modify the shock magnitude and waveform. Then, MATLAB was used to program a graphical user interface (GUI) for the waveform prediction simulator, to verify the validity of the proposed system. Finally, the effectiveness of the proposed simulator-featuring controllable MR damper is demonstrated by comparing the simulated and experimental results.

## 2. Force-Velocity Diagram Modeling of the High Capacity Shock MR Damper

### 2.1. Double Pulse Waveform Shock Test Setup

Since the full-scale submarine or naval vessel testing for heavy-weight components requires a lot of time and cost, BV-43/85 (developed in Germany) rules that a double half-sine acceleration profile, which can represent the shock loading caused by an underwater explosion, is applied to shipboard or submarine components [25,26]. As shown in Figure 2, the acceleration profile consists of three parts. In BV-43/85, after first half-sine acceleration waveform (Part I) strikes the test mass, second negative half-sine acceleration waveform (Part II) and very small residual vibration should be applied to test mass (Part III). In particular, the components are mounted on test mass. In order to dissipate the impact energy and realize the required acceleration waveform profile, MR damper with large power dissipation capacity is utilized. After striking, the impact mass is moved in the opposite direction to the test mass due to the elastic energy of programmer and air spring. Also, when impact mass is decoupled with test mass, the impact mass ideally moves at constant velocity. The positive half-wave (Part I) shows the effect of the magnitude of the impact, and the negative half-wave (Part II) and Part III are related to the energy dissipation system. In Figure 2, the magnitude of the acceleration in Part II is lower than that in Part I, and the magnitude of the residual vibration (Part III) is zero. Figure 3 shows a schematic diagram of the shock test experimental setup used in this study. The set up comprises a hydraulic velocity generator, impact mass, shock spring programmer, test mass, and MR damper. The specimen is attached to the test mass. In the impact-testing machine, the hydraulic velocity generator accelerates the impact mass, which then strikes the programmer shock spring and test mass with a uniform motion. The high-impact load and velocity are then applied to the MR damper, which is located between the test mass and a fixed wall. The MR damper is used to reduce the transmitted reaction force (Part II) and residual vibration (Part III). In other words, the generation of a shock-wave profile, particularly for Parts II and III, is the primary function of the MR damper. 

The horizontal hydraulic double pulse shock tester is composed of the velocity generator, impact mass, shock programmer, test table, and MR damper. The shock test process was as follows. First, the velocity generator accelerates the impact mass at a specified velocity. The impact mass, accelerated to the maximum velocity, travels at constant velocity and impacts the shock programmer installed in front of the test table. After the impact, the impact mass bounces off and the impact acceleration of the test table is altered by the MR damper, located between the test table and the fixed wall. The impact acceleration is measured by an accelerometer and load cell, and the internal pressure of the MR damper is measured using a pressure sensor. The impact acceleration of the test table in this structure typically has the following two waveform types: the primary waveform, with a short period and positive acceleration value due to the impact between the accelerated impact mass and the test table, and the secondary impact acceleration, which is altered by the MR damper after the primary waveform. Figure 4 shows a photo of the MR damper and test equipment.

### 2.2. Double Pulse Shock Test Results and Analysis

To investigate the dynamic behavior and impact acceleration variation properties of the MR damper at high velocity and high mass, impact experiments were conducted using the horizontal hydraulic double pulse shock test equipment. Depending on the objective of the experiment, the damping force tuning properties of MR damper were evaluated by varying the current applied to the MR damper while fixing the impact mass, impact velocity, shock programmer stiffness, and test table mass. The damping force produced by the MR damper was plotted in the time domain and analyzed in the frequency domain, as shown in Figure 5. When high-mass structures impact at high velocity, various structural vibrations can be plotted simultaneously with major curves, as shown in Figure 6, Figure 7, Figure 8 and Figure 9. The aim of this experiment was to obtain the force-velocity (F-V) diagram of the MR damper. However, when structural vibrations are mixed together in this process, the accurate modeling becomes difficult because a clear graph cannot be obtained. Thus, the structural vibration signal in the high frequency range, beyond that of the major waveform, needs to be removed. To accomplish this, a hamming window with a cut-off frequency of 500 Hz and a sampling frequency of 5120 Hz, and a finite impulse-response (FIR) low-pass filter were used for signal processing. The order of FIR filer is 100. The issue of time delay from the higher order was resolved and reorganized based on the impact velocity, and the final results are shown in Figure 6, Figure 7, Figure 8 and Figure 9 and Table 1. The impact and test table masses were 140 and 280 kg, respectively, and the stiffness of the shock programmer was 4.6 MN/m. The results of the damping force of the MR damper in the time domain were measured using a load cell placed between the test table and the MR damper piston rod end. The results of the piston rod velocity in the time domain were measured using an accelerometer installed on the table mass. F-V curve results were obtained by plotting the measured damping force in the y-axis and the velocity in the x-axis. 

The following observations were made from the shock experiment results. As the impact velocity increased, the damping force due to the viscous term of the fluid, or that of the MR damper at 0 A, increased proportionally, but the ratios of the magnetic field to the controllable damping force due to the MR yield stress and that due to the viscous term decreased. As the velocity increased, the effect of the damping force due to the magnetic field decreased, because the duration of the fluid flow through the orifice within the MR damper was shorter than the time required to observe the MR effect. Overall, the damping force increased but the controllability decreased.

### 2.3. Damping F-V Curve Modeling

The magnetorheological fluid is the suspension of iron particles dispersed in a non-magnetic medium. The behavior of the MR fluid without magnetic field are similar with that of Newtonian fluid, but the unrestrained particles transform to a chain-like structuring under an external magnetic field. A relationship between magnetic input and shear stress of MR fluid has been researched and established as follows [27,28]:(1)$$\tau =\eta \dot{\gamma }+\tau_{y}\left(B\right)$$
where τ and η are the total shear stress and viscosity constant of the MR fluid, respectively. $\dot{\gamma }$ is the relative shear rate. $\tau_{y}\left(B\right)$ is the dynamic yield stress of the MR fluid, which is a function of the magnetic field intensity. To generate magnetic field, a DC current input is applied to coil. Accordingly, the total shear stress of MR fluid is affected by the current input and relative shear rate. Especially, some research reported that the viscosity can be tuned by the magnetic field intensity and magnetic field applying time for nanoparticle-added MR fluid [29,30]. Then, the damping force of MR damper is determined by the total shear stress of MR fluid and geometric configuration. However, when big shock is given to MR damper, the response of MR damper is different with that of MR damper with relative low piston velocity. Additionally, the shear rate of MR fluid is dependent on the piston velocity of MR damper. It is judged that the relationship between dynamic yield stress, shear rate and total shear stress is nonlinear. So, to predict the response of MR damper, an empirical formula is constructed in this work. The experimental results are divided into intervals with different velocity ranges, and a linear function modeling based on piston velocity and current input is conducted for each interval. The empirical formula can be utilized as the command law for generating varied double pulse waveform.

Figure 10 shows the F-V curve obtained in Test 4, divided into intervals for modeling. It can be seen from Figure 10 that the F-V diagram obtained from the experiments are significantly nonlinear. Because it is difficult to model this with one equation, the method of modeling intervals was adopted. The velocity intervals were 0.0, 1.0, 1.8, 2.0, 2.5, 3.25, 3.5, and 4.5 m/s based on the extent of the F-V curve slope variation. Each interval was expressed with a linear function of the velocity. A simplified form of the function is: (2)F(v)=c(i)v+d(i) where, c(i) and d(i) are coefficients that change according to the input current, and F(v) is the damping force of the MR damper, which varies according to the velocity ν and the input current. Table 2 shows the c(i) and d(i) values obtained. Using the coefficients shown in Table 2, depending on whether the velocity is positive or negative, the damping force can be expressed as follows:(3)if vel≥0, F(v)=c(i)(v−vel.)+d(i)
(4)if vel≤0, F(v)=−c(i)(−v−vel.)−d(i)

Figure 11 shows the MR damper damping force curve modeling result obtained from the above equations. Figure 11a shows a comparison of the experiment results, divided into intervals, with the modeling results; the graph on the right shows the damping force results obtained using Equations (3) and (4) with input velocities between −4.5 m/s and 4.5 m/s. In Figure 11a, the solid black line is the curve obtained from the modeling, and the other color-lines are obtained from the experiments. As shown in Figure 11a, the simulation and experimental curves are very similar. Figure 11b shows a continuous curve after checking that the next modeling value and the experiment value matched. A linear function was used from −1 to 1 m/s because hysteresis characteristics in the low velocity range were not considered, as the goal was to express the high velocity range using the MR damper modeling. Moving on to the maximum velocity portion, it can be observed that, similar to the trend in the experimental result, the controllable range was approximately constant with increasing velocity, but the minimum damping force increased proportionally, indicating a lower control performance.

## 3. Waveform Prediction Simulator

A waveform prediction simulator is a program that calculates the input current considering the characteristics of the MR damper and the parameter values of the shock system when there is a desired waveform in a large-scale high-mass impact system. To construct such a program, the dynamics equations of the impact system have to be calculated and the MR damper model has to be applied. Finally, the system has to be programmed.

### 3.1. Dynamic Modeling of Dual Shock Waveform: Primary Shock Waveform

When the impact mass strikes the test table, a primary shock and vibration are generated. With air as the spring, corresponding to the dummy mass (*m*_3_), the governing equation of the primary shock waveform can be derived as follows: (5)$$\begin{array}{l} m_{1}{\ddot{x}}_{1}+k_{1}x_{1}-k_{1}x_{2}=0\\ m_{2}{\ddot{x}}_{2}-k_{1}x_{1}+\left(k_{1}+k_{2}\right)x_{2}-k_{2}x_{3}=0\\ m_{3}{\ddot{x}}_{3}-k_{2}x_{2}+k_{2}x_{3}=0 \end{array}$$ Because *x*_1_, *x*_2_, and *x*_3_ are the sum of sinusoidal waves, Equation (5) can be transformed as follows: (6)$$\left[\begin{array}{ccc} \left(k_{1}-m_{1}\omega^{2}\right) & -k_{1} & 0\\ -k_{1} & \left(k_{1}+k_{2}-m_{2}\omega^{2}\right) & -k_{2}\\ 0 & -k_{2} & \left(k_{2}-m_{3}\omega^{2}\right) \end{array}\right]\left\{\begin{array}{c} X_{1}\\ X_{2}\\ X_{3} \end{array}\right\}=\left\{\begin{array}{c} 0\\ 0\\ 0 \end{array}\right\}$$ From the determinant of Equation (6), the natural frequencies are obtained as follows: (7)$$\omega_{1}=0$$
(8)$$\omega_{2}=\frac{k_{2}m_{1}m_{2}+k_{1}m_{1}m_{3}+k_{2}m_{1}m_{3}+k_{1}m_{2}m_{3}+\sqrt{4m_{1}m_{2}m_{3}(-k_{1}k_{2}m_{1}-k_{1}k_{2}m_{2}-k_{1}k_{2}m_{3})+{\left(k_{2}m_{1}m_{2}+k_{1}m_{1}m_{3}+k_{2}m_{1}m_{3}+k_{1}m_{2}m_{3}\right)}^{2}}}{2m_{1}m_{2}m_{3}}$$
(9)$$\omega_{3}=\frac{k_{2}m_{1}m_{2}+k_{1}m_{1}m_{3}+k_{2}m_{1}m_{3}+k_{1}m_{2}m_{3}-\sqrt{4m_{1}m_{2}m_{3}(-k_{1}k_{2}m_{1}-k_{1}k_{2}m_{2}-k_{1}k_{2}m_{3})+{\left(k_{2}m_{1}m_{2}+k_{1}m_{1}m_{3}+k_{2}m_{1}m_{3}+k_{1}m_{2}m_{3}\right)}^{2}}}{2m_{1}m_{2}m_{3}}$$ Then, the general solution of Equation (5) can be expressed as follows:(10)$$\begin{array}{l} x_{1}(t)=X_{1}^{\left(1\right)}\cos(\omega_{1}t+\varphi_{1})+X_{1}^{\left(2\right)}\cos(\omega_{2}t+\varphi_{2})+X_{1}^{\left(3\right)}\cos(\omega_{3}t+\varphi_{3})\\ x_{2}(t)=r_{12\_1}X_{1}^{\left(1\right)}\cos(\omega_{1}t+\varphi_{1})+r_{12\_2}X_{1}^{\left(2\right)}\cos(\omega_{2}t+\varphi_{2})+r_{12\_3}X_{1}^{\left(3\right)}\cos(\omega_{3}t+\varphi_{3})\\ x_{3}(t)=r_{13\_1}X_{1}^{\left(1\right)}\cos(\omega_{1}t+\varphi_{1})+r_{13\_2}X_{1}^{\left(2\right)}\cos(\omega_{2}t+\varphi_{2})+r_{13\_3}X_{1}^{\left(3\right)}\cos(\omega_{3}t+\varphi_{3}) \end{array}$$
(11)$$\begin{array}{l} where\\ r_{12\_1}=\frac{\left(k_{1}-m_{1}\omega_{1}{}^{2}\right)}{k_{1}},\;r_{23\_1}=\frac{k_{2}}{\left(k_{2}-m_{3}\omega_{1}{}^{2}\right)},\;r_{13\_1}=r_{12\_1}\times r_{23\_1}=\frac{k_{2}(k_{1}-m_{1}\omega_{1}{}^{2})}{k_{1}(k_{2}-m_{3}\omega_{1}{}^{2})},\\ r_{12\_2}=\frac{\left(k_{1}-m_{1}\omega_{2}{}^{2}\right)}{k_{1}},\;r_{23\_2}=\frac{k_{2}}{\left(k_{2}-m_{3}\omega_{2}{}^{2}\right)},\;r_{13\_2}=r_{12\_2}\times r_{23\_2}=\frac{k_{2}(k_{1}-m_{1}\omega_{2}{}^{2})}{k_{1}(k_{2}-m_{3}\omega_{2}{}^{2})},\\ r_{12\_3}=\frac{\left(k_{1}-m_{1}\omega_{3}{}^{2}\right)}{k_{1}},\;r_{23\_3}=\frac{k_{2}}{\left(k_{2}-m_{3}\omega_{3}{}^{2}\right)},\;r_{13\_3}=r_{12\_3}\times r_{23\_3}=\frac{k_{2}(k_{1}-m_{1}\omega_{3}{}^{2})}{k_{1}(k_{2}-m_{3}\omega_{3}{}^{2})} \end{array}$$ By using the initial conditions, Equation (5) can be re-written as follows:(12)$$\begin{array}{l} x_{1}(0)=0=X_{1}^{\left(1\right)}\cos\varphi_{1}+X_{1}^{\left(2\right)}\cos\varphi_{2}+X_{1}^{\left(3\right)}\cos\varphi_{3}\\ {\dot{x}}_{1}(0)=v_{impact}=-\omega_{1}X_{1}^{\left(1\right)}\sin\varphi_{1}-\omega_{2}X_{1}^{\left(2\right)}\sin\varphi_{2}-\omega_{3}X_{1}^{\left(3\right)}\sin\varphi_{3}\\ x_{2}(0)=0=r_{12\_1}X_{1}^{\left(1\right)}\cos\varphi_{1}+r_{12\_2}X_{1}^{\left(2\right)}\cos\varphi_{2}+r_{12\_3}X_{1}^{\left(3\right)}\cos\varphi_{3}\\ {\dot{x}}_{2}(0)=0=-\omega_{1}r_{12\_1}X_{1}^{\left(1\right)}\sin\varphi_{1}-\omega_{2}r_{12\_2}X_{1}^{\left(2\right)}\sin\varphi_{2}-\omega_{3}r_{12\_3}X_{1}^{\left(3\right)}\sin\varphi_{3}\\ x_{3}(0)=0=r_{13\_1}X_{1}^{\left(1\right)}\cos\varphi_{1}+r_{13\_2}X_{1}^{\left(2\right)}\cos\varphi_{2}+r_{13\_3}X_{1}^{\left(3\right)}\cos\varphi_{3}\\ {\dot{x}}_{3}(0)=0=-\omega_{1}r_{13\_1}X_{1}^{\left(1\right)}\sin\varphi_{1}-\omega_{2}r_{13\_2}X_{1}^{\left(2\right)}\sin\varphi_{2}-\omega_{3}r_{13\_3}X_{1}^{\left(3\right)}\sin\varphi_{3} \end{array}$$ From Equation (12), the following angle value is obtained: (13)$$\left\{\begin{array}{c} \varphi_{1}\\ \varphi_{2}\\ \varphi_{3} \end{array}\right\}=\left\{\begin{array}{c} \frac{\pi }{2}\\ \frac{\pi }{2}\\ \frac{\pi }{2} \end{array}\right\}$$ Finally, the particular solution of Equation (5) is obtained from Equations (10) and (13): (14)$$\begin{array}{l} x_{1}(t)=-X_{1}^{\left(1\right)}\sin\omega_{1}t-X_{1}^{\left(2\right)}\sin\omega_{2}t-X_{1}^{\left(3\right)}\sin\omega_{3}t\\ {\dot{x}}_{1}(t)=-\omega_{1}X_{1}^{\left(1\right)}\cos\omega_{1}t-\omega_{2}X_{1}^{\left(2\right)}\cos\omega_{2}t-\omega_{3}X_{1}^{\left(3\right)}\cos\omega_{3}t\\ {\ddot{x}}_{1}(t)=\omega_{1}{}^{2}X_{1}^{\left(1\right)}\sin\omega_{1}t+\omega_{2}{}^{2}X_{1}^{\left(2\right)}\sin\omega_{2}t+\omega_{3}{}^{2}X_{1}^{\left(3\right)}\sin\omega_{3}t \end{array}$$
(15)$$\begin{array}{l} x_{2}(t)=-r_{12\_1}X_{1}^{\left(1\right)}\sin\omega_{1}t-r_{12\_2}X_{1}^{\left(2\right)}\sin\omega_{2}t-r_{12\_3}X_{1}^{\left(3\right)}\sin\omega_{3}t\\ {\dot{x}}_{2}(t)=-r_{12\_1}\omega_{1}X_{1}^{\left(1\right)}\cos\omega_{1}t-r_{12\_2}\omega_{2}X_{1}^{\left(2\right)}\cos\omega_{2}t-r_{12\_3}\omega_{3}X_{1}^{\left(3\right)}\cos\omega_{3}t\\ {\ddot{x}}_{2}(t)=r_{12\_1}\omega_{1}{}^{2}X_{1}^{\left(1\right)}\sin\omega_{1}t+r_{12\_2}\omega_{2}{}^{2}X_{1}^{\left(2\right)}\sin\omega_{2}t+\omega_{3}{}^{2}r_{12\_3}X_{1}^{\left(3\right)}\sin\omega_{3}t \end{array}$$
(16)$$\begin{array}{l} x_{3}(t)=-r_{13\_1}X_{1}^{\left(1\right)}\sin\omega_{1}t-r_{13\_2}X_{1}^{\left(2\right)}\sin\omega_{2}t-r_{13\_3}X_{1}^{\left(3\right)}\sin\omega_{3}t\\ {\dot{x}}_{3}(t)=-r_{13\_1}\omega_{1}X_{1}^{\left(1\right)}\cos\omega_{1}t-r_{13\_2}\omega_{2}X_{1}^{\left(2\right)}\cos\omega_{2}t-r_{13\_3}\omega_{3}X_{1}^{\left(3\right)}\cos\omega_{3}t\\ {\ddot{x}}_{3}(t)=r_{13\_1}\omega_{1}{}^{2}X_{1}^{\left(1\right)}\sin\omega_{1}t+r_{13\_2}\omega_{2}{}^{2}X_{1}^{\left(2\right)}\sin\omega_{2}t+\omega_{3}{}^{2}r_{13\_3}X_{1}^{\left(3\right)}\sin\omega_{3}t \end{array}$$

Because the impact mass is decoupled after striking the test mass and $\omega_{2}$ is very low, the frequency of the primary shock wave form is determined approximately by $\omega_{3}$. Accordingly, the shock duration time and maximum acceleration of the primary shock waveform can be expressed as follows:(17)$$Shock\;duration\;time=\frac{\pi }{\omega_{3}}$$
(18)$$Max.\;Acc.={\ddot{x}}_{2}(T)=r_{12\_2}\omega_{2}{}^{2}X_{1}^{\left(2\right)}\sin\omega_{2}T+\omega_{3}{}^{2}r_{12\_3}X_{1}^{\left(3\right)}\sin\omega_{3}T\;\left(T=\frac{\pi }{2\omega_{3}}\right)$$

### 3.2. Dynamic Modeling of Dual Shock Waveform: Secondary Shock Waveform

A similar approach is applied to derive the dynamic model of the secondary shock waveform. The difference between the primary and secondary waveforms is the initial conditions. From Equation (16) with $T=\pi /\omega_{3}$, the initial conditions of the secondary waveform can be expressed as follows: (19)$$\begin{array}{l} D_{2init.}=-r_{12\_1}X_{1}^{\left(1\right)}\sin\omega_{1}T-r_{12\_2}X_{1}^{\left(2\right)}\sin\omega_{2}T-r_{12\_3}X_{1}^{\left(3\right)}\sin\omega_{3}T\\ V_{2init.}=-r_{12\_1}\omega_{1}X_{1}^{\left(1\right)}\cos\omega_{1}T-r_{12\_2}\omega_{2}X_{1}^{\left(2\right)}\cos\omega_{2}T-r_{12\_3}\omega_{3}X_{1}^{\left(3\right)}\cos\omega_{3}T \end{array}$$
(20)$$\begin{array}{l} D_{3init.}=-r_{13\_1}X_{1}^{\left(1\right)}\sin\omega_{1}T-r_{13\_2}X_{1}^{\left(2\right)}\sin\omega_{2}T-r_{13\_3}X_{1}^{\left(3\right)}\sin\omega_{3}T\\ V_{3init.}=-r_{13\_1}\omega_{1}X_{1}^{\left(1\right)}\cos\omega_{1}T-r_{13\_2}\omega_{2}X_{1}^{\left(2\right)}\cos\omega_{2}T-r_{13\_3}\omega_{3}X_{1}^{\left(3\right)}\cos\omega_{3}T \end{array}$$ where $D_{2init.},\;V_{2init.},\;D_{3init.},$ and $V_{3init.}$ are the initial displacement and velocity of the second waveform for test mass (*m*_2_) and dummy mass (*m*_3_), respectively. The governing equation of the secondary shock waveform can be derived as follows:(21)$$\begin{array}{l} m_{2}{\ddot{x}}_{2}+k_{2}x_{2}-k_{2}x_{3}=0\\ m_{3}{\ddot{x}}_{3}-k_{2}x_{2}+k_{2}x_{3}=0 \end{array}$$ Equation (21) can be transformed as follows:(22)$$\left[\begin{array}{cc} \left(k_{2}-m_{2}\omega^{2}\right) & -k_{2}\\ -k_{2} & \left(k_{2}-m_{3}\omega^{2}\right) \end{array}\right]\left\{\begin{array}{c} X_{2}\\ X_{3} \end{array}\right\}=\left\{\begin{array}{c} 0\\ 0 \end{array}\right\};\;\omega_{1}=0,\;\omega_{2}=\sqrt{\frac{k_{2}(m_{2}+m_{3})}{m_{2}m_{3}}}$$ Then, the general solution of Equation (21) can be expressed as follows:(23)$$\begin{array}{l} x_{2}(t)=X_{2}^{\left(1\right)}\cos(\omega_{1}t+\varphi_{1})+X_{2}^{\left(2\right)}\cos(\omega_{2}t+\varphi_{2})\\ x_{3}(t)=r_{1}X_{2}^{\left(1\right)}\cos(\omega_{1}t+\varphi_{1})+r_{2}X_{2}^{\left(2\right)}\cos(\omega_{2}t+\varphi_{2})\\ where\\ r_{1}=\frac{X_{3}{}^{\left(1\right)}}{X_{2}{}^{\left(1\right)}}=\frac{-m_{2}\omega_{1}^{2}+k_{2}}{k_{2}}=1,\;r_{2}=\frac{X_{3}{}^{\left(2\right)}}{X_{2}{}^{\left(2\right)}}=\frac{-m_{2}\omega_{2}^{2}+k_{2}}{k_{2}}=-\frac{m_{2}}{m_{3}} \end{array}$$ From the initial conditions, Equations (19) and (20), the following equations can be:(24)$$\begin{array}{l} x_{2}(0)=X_{2}^{\left(1\right)}\cos\phi_{1}+X_{2}^{\left(2\right)}\cos\phi_{2}=D_{2init.}\\ {\dot{x}}_{2}(0)=-\omega_{1}X_{2}^{\left(1\right)}\sin\phi_{1}-\omega_{2}X_{2}^{\left(2\right)}\sin\phi_{2}=V_{2init.}\\ x_{3}(0)=r_{1}X_{2}^{\left(1\right)}\cos\phi_{1}+r_{2}X_{2}^{\left(2\right)}\cos\phi_{2}=D_{3init.}\\ {\dot{x}}_{3}(0)=-\omega_{1}r_{1}X_{2}^{\left(1\right)}\sin\phi_{1}-\omega_{2}r_{2}X_{2}^{\left(2\right)}\sin\phi_{2}=V_{3init.} \end{array}$$ From Equation (24), the expression for the angle is obtained:(25)$$\phi_{1}=\tan^{-1}\left(\frac{-r_{2}V_{2init.}+V_{3init.}}{\omega_{1}(r_{2}D_{2init.}-D_{3init.})}\right)=\frac{\pi }{2},\;\phi_{2}=\tan^{-1}\left(\frac{r_{1}V_{2init.}-V_{3init.}}{\omega_{2}(-r_{1}D_{2init.}+D_{3init.})}\right)$$ Finally, the particular solution of Equation (21) is obtained from Equations (23) and (25):(26)$$\begin{array}{l} x_{2}(t)=X_{2}^{\left(2\right)}\cos(\omega_{2}t+\phi_{2})\\ {\dot{x}}_{2}(t)=-\omega_{2}X_{2}^{\left(2\right)}\sin(\omega_{2}t+\phi_{2})\\ {\ddot{x}}_{2}(t)=-\omega_{2}{}^{2}X_{2}^{\left(2\right)}\cos(\omega_{2}t+\phi_{2}) \end{array}$$
(27)$$\begin{array}{l} x_{3}(t)=r_{2}X_{2}^{\left(2\right)}\cos(\omega_{2}t+\phi_{2})\\ {\dot{x}}_{3}(t)=-\omega_{2}X_{2}^{\left(2\right)}\sin(\omega_{2}t+\phi_{2})\\ {\ddot{x}}_{3}(t)=-\omega_{2}{}^{2}X_{2}^{\left(2\right)}\cos(\omega_{2}t+\phi_{2}) \end{array}$$

Because the first natural frequency of the secondary waveform is zero, the frequency of the secondary shock wave form is $\omega_{2}$. Accordingly, the shock duration time and maximum acceleration of the secondary shock waveform can be expressed as follows:(28)$$Shock\;duration\;time=\frac{\pi }{\omega_{2}}$$
(29)$$Max.Acc.={\ddot{x}}_{2}(T)=-\omega_{2}{}^{2}X_{2}^{\left(2\right)}\cos(\omega_{2}T+\phi_{2})\;\left(T=\frac{\pi }{2\omega_{2}}\right)$$

### 3.3. Waveform Prediction Simulator Program Structure

The MATLAB GUI program was used to produce the waveform prediction simulator. In the structure, the equations for the primary and secondary waveforms analyzed in Section 3.1 and Section 3.2 were all input and solved to obtain the required values. The procedure is described in the following paragraph. The impact and test table masses were fixed and the material properties of the variables that could be obtained analytically were determined, including the impact velocity, dummy mass, shock programmer stiffness, and air spring stiffness. To integrate these into the MR damper model, the values were changed based on the obtained dummy mass. The waveform properties were determined by solving the response of the entire impact system, including the MR damper model, using the Runge Kutta method, based on the following equations:(30)$$\begin{array}{l} m_{2}{\ddot{x}}_{2}+k_{2}x_{2}-k_{2}x_{3}=0\\ m_{3}{\ddot{x}}_{3}-k_{2}x_{2}+k_{2}x_{3}-{\left(F(v)\right)}_{\mathrm{MR}}=0 \end{array}$$

Multiple cases that yielded the desired waveform were identified and that with the most reasonable variables was selected.

Figure 12 shows the proposed GUI program and flow chart. First, the waveform properties of the impact duration and maximum impact acceleration for the primary and secondary waveforms are input, followed by the setting of the impact, test, and dummy mass ranges for the system. In <1>, the variables that can realize first shock waveform are determined, except for the MR damper. Based on this, the numerical analysis, including the MR damper, is carried out; the various cases that approach the waveform properties are listed in process <2>. Among the listed cases, the proper one is manually selected and the predicted waveform is calculated. The secondary waveform time and acceleration, where the residual vibrations are attenuated, are expressed according to the damping force model of the MR damper (Equations (2)–(4)), in process <3>.

Figure 13 and Figure 14 show the experimental and simulation results obtained with the waveform properties period and maximum acceleration set to 0.01 s and 590 m/s^2^ (60G), respectively, for the primary waveform and 0.06 s for the period of secondary waveform. The parameter values obtained through the waveform prediction program were used to carry out the simulation. From the simulator, the masses of impact, test table, and dummy were determined to be 140, 200, and 100 kg, respectively. The stiffness of programmer shock spring and air spring were 4.6 MN/m and 0.39 MN/m, respectively. The magnitude of the impact velocity was 6.1 m/s. The simulation results were found to be in relatively good agreement with the experimental results. From the simulation experimental results, it can be inferred that the proposed dynamic modeling of the test system and experimental damping force model of MR damper are effective for predicting shockwave form. From Figure 13 and Figure 14, the magnitude and duration time of the second shockwave are increased according to current input. This is why the reaction force of the MR damper is proportional to the damping force of the MR damper. Also, the dynamic energy should be dissipated during part III, owing to the MR damper. In particular, the dynamic energy of the testing system is proportional to the acceleration of test mass. From Figure 14, it can be known that the Part III acceleration results for 0.5 A is reduced from 84.15 m/s^2^ to 43.87 m/s^2^ in 0.149 s.

However, the measured magnitude of the first shockwave is different from that of the predicted shockwave. During the impact between impact and test mass, the high-impact velocity is unequally applied to test mass. This is the reason for the difference at the first shockwave. The maximum error value of the first shockwave is 173.14 m/s^2^ at 0 A. The other tangible difference is the duration time of the second shockwave (t_2_). The maximum difference value of the second shockwave is 0.01 s at 0 A. This is related with the nonlinearity of the elastic and damping force for sudden impact and slow response time of the MR damper. In addition, the structure of the proposed program involves determining the value by varying the dummy mass, which sometimes becomes too large, resulting in a lower MR damper efficiency. Thus, in future studies, the waveform should be found by varying other material properties while leaving the maximum mass unchanged.

## 4. Conclusions

In this study, we proposed an MR damper modeling and a waveform prediction program to simulate the double pulse from an underwater explosion. Impact experiments were carried out to obtain the damping F-V curve, and that with the highest velocity was modeled mathematically. For this, the curve with high nonlinearity was divided into intervals with different velocity ranges, with rapid variations in the damping force. A linear function modeling was conducted for each interval. Then, the simulation results were compared with the actual impact experiment results. From the comparison results between simulation and experimental results, the proposed dynamic modeling of the system and damping are quite self-explanatory, showing good agreement with experimental results. In addition, it has been confirmed that MR damper featuring sphere-like iron particle shows a high damping force up to 200 kN. Since the performance on damping force, durability, sedimentation, and dispersible stability of MR fluid can be varied according to the shape of particles, the performance of MR dampers with different particle shapes will be evaluated as a second phase of this work in the future.

## Figures and Tables

**Figure 1 ijms-21-01149-f001:**
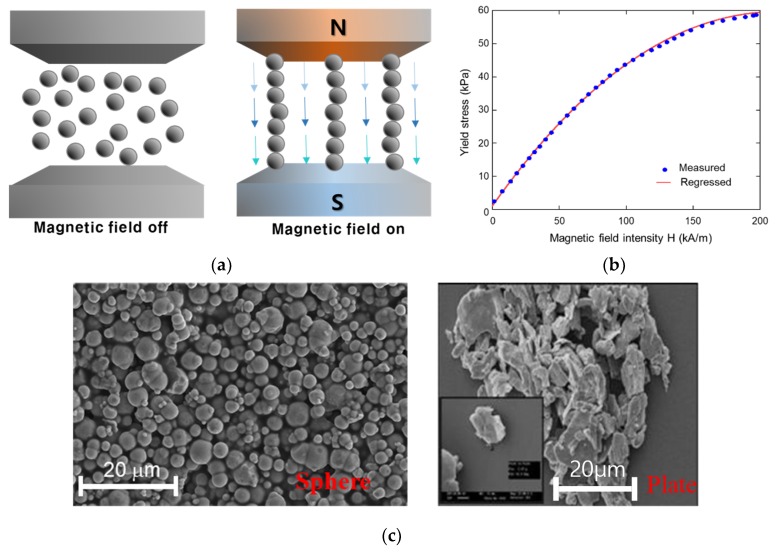
MR fluid: (**a**) Phenomenological characteristics; (**b**) Yield stress results; (**c**) Shape of particle.

**Figure 2 ijms-21-01149-f002:**
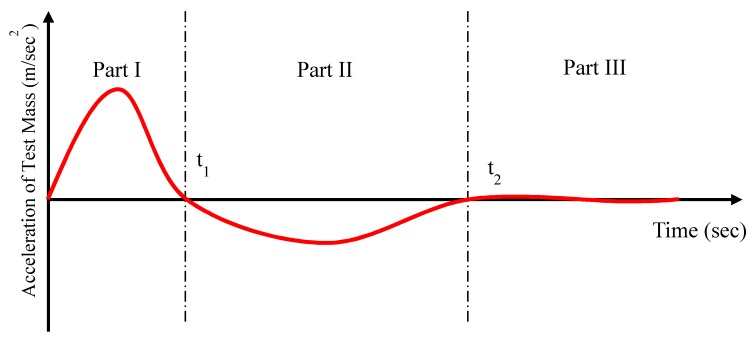
Shape of the double pulse waveform profile.

**Figure 3 ijms-21-01149-f003:**
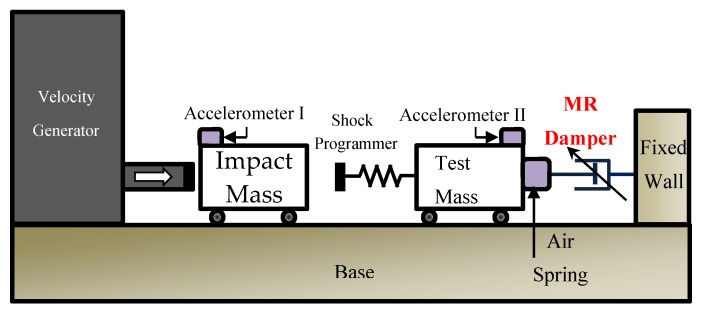
Schematic diagram of the horizontal hydraulic double pulse shock test setup.

**Figure 4 ijms-21-01149-f004:**
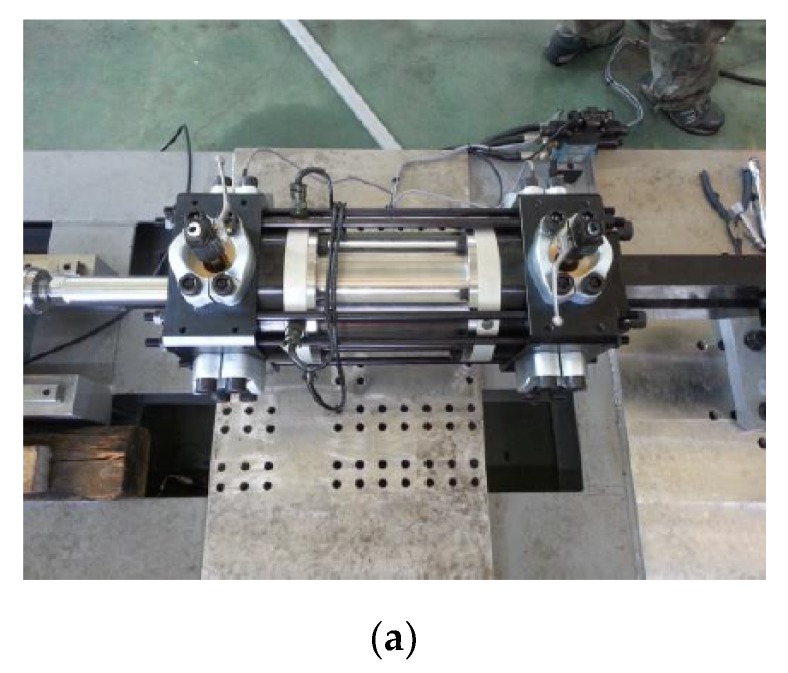
Photograph of the double pulse shock test setup: (**a**) Manufacture MR damper; (**b**) Experimental apparatus.

**Figure 5 ijms-21-01149-f005:**
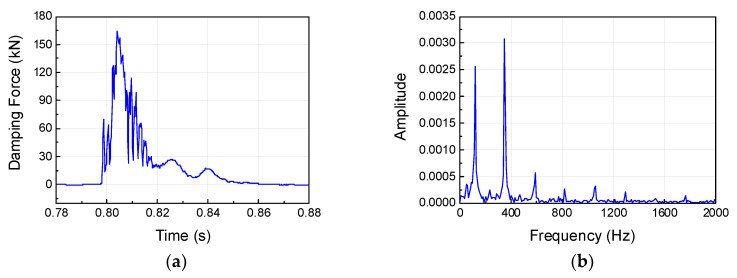
Measured damping force in the time and frequency domains: (**a**) Measured damping force in the time domain; (**b**) Measured damping force in the frequency domain.

**Figure 6 ijms-21-01149-f006:**
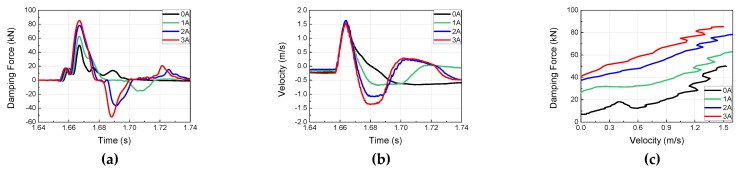
MR damper experimental results at an impact velocity of 2.5 m/s (Test 1): (**a**) Damping force Piston velocity; (**b**) Piston Velocity; (**c**) F-V diagram of MR damper.

**Figure 7 ijms-21-01149-f007:**
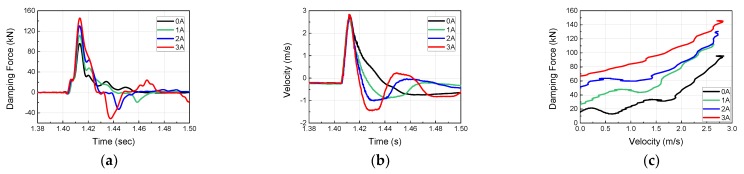
MR damper experimental results at an impact velocity of 5.37 m/s (Test 2): (**a**) Damping force Piston velocity; (**b**) Piston Velocity; (**c**) F-V diagram of MR damper.

**Figure 8 ijms-21-01149-f008:**
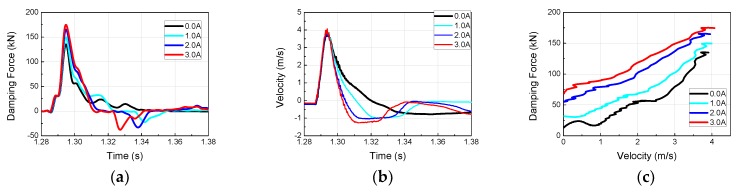
MR damper experimental results at an impact velocity of 7.75 m/s (Test 3): (**a**) Damping force Piston velocity; (**b**) Piston Velocity; (**c**) F-V diagram of MR damper.

**Figure 9 ijms-21-01149-f009:**
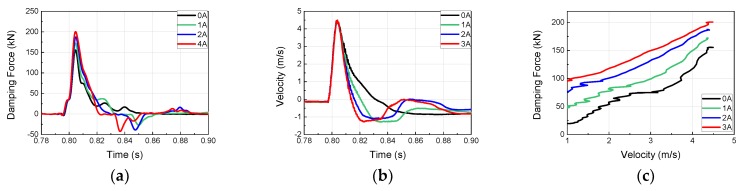
MR damper experimental results at an impact velocity of 9.12 m/s (Test 4): (**a**) Damping force; (**b**) Piston velocity; (**c**) force-velocity (F-V) diagram of MR damper.

**Figure 10 ijms-21-01149-f010:**
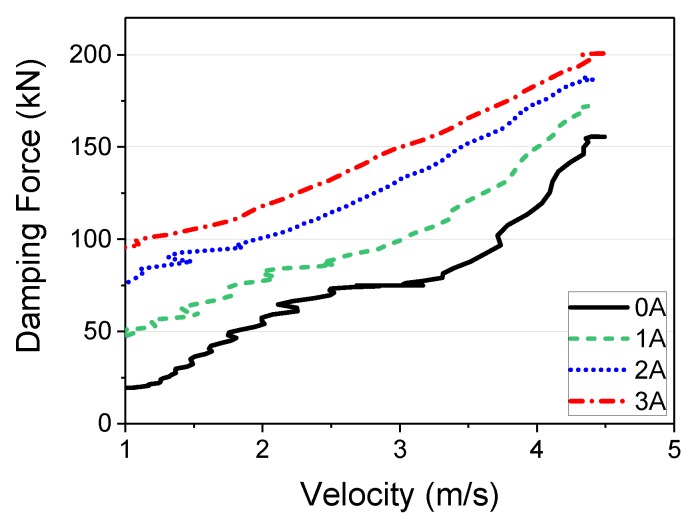
F-V curve (Test. 4).

**Figure 11 ijms-21-01149-f011:**
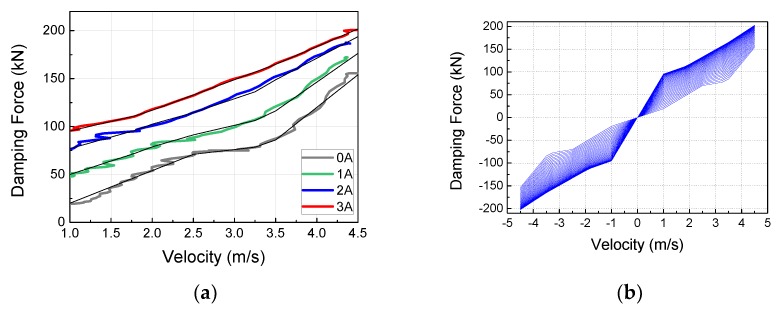
Estimation results of F-V model for MR damper: (**a**) Comparison results between experimental and simulator results; (**b**) Simulation F-V curve according to continuous current inputs.

**Figure 12 ijms-21-01149-f012:**
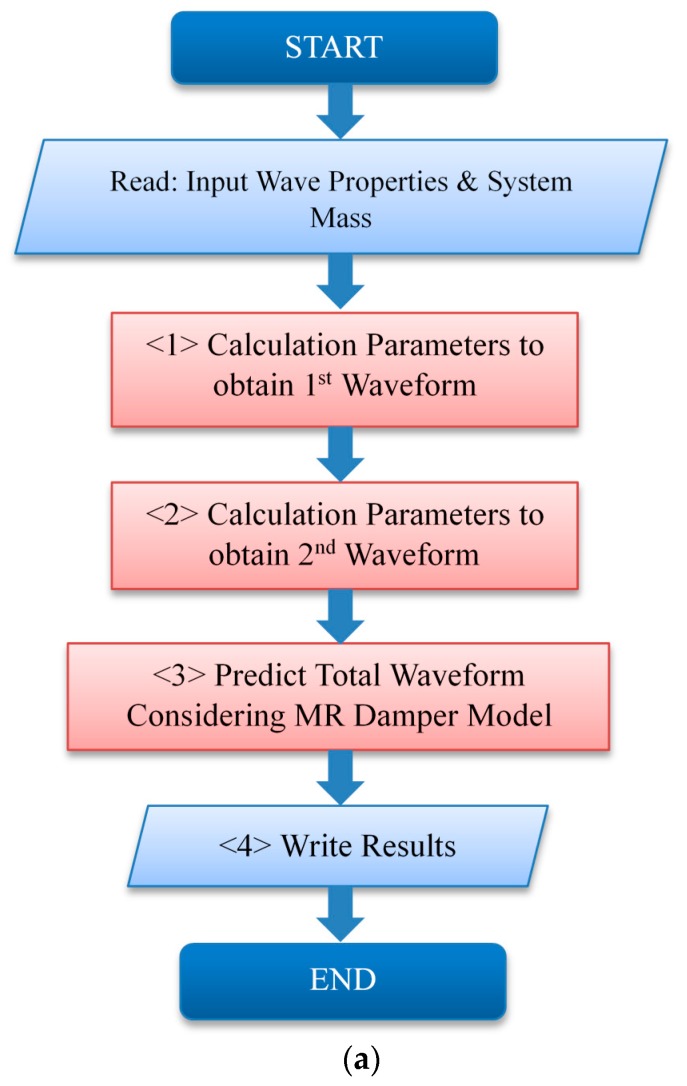
GUI of the waveform prediction simulator program: (**a**)Flow chart; (**b**) GUI program.

**Figure 13 ijms-21-01149-f013:**
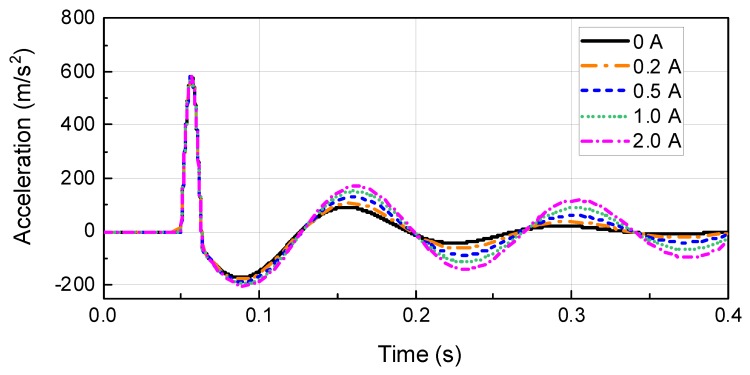
Simulation result of the waveform prediction program.

**Figure 14 ijms-21-01149-f014:**
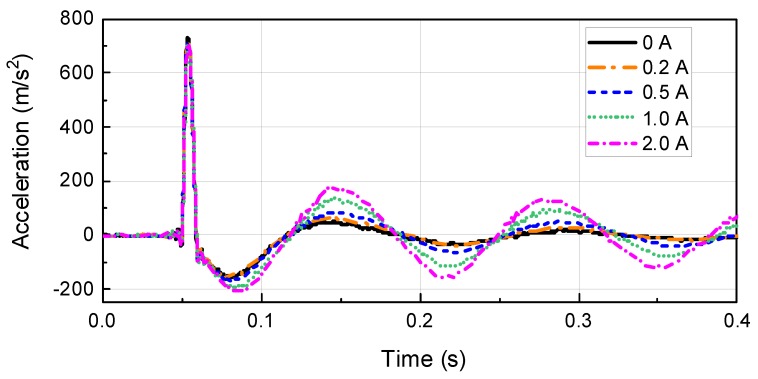
Experimental result of the waveform prediction program.

**Table 1 ijms-21-01149-t001:** Measured experimental results.

Test No.	Impact Force at Test Table	Damping Force		Max. Velocity of Piston	Dynamic Range of Damping Force
		Max.	Min.		
1	108.208 kN	84.0 kN	48.9 kN	1.5 m/s	1.72
2	183.708 kN	144.0 kN	97.9 kN	2.75 m/s	1.47
3	246.678 kN	177.7 kN	132.5 kN	4.2 m/s	1.341
4	301.194 kN	201.3 kN	150.7 kN	4.4 m/s	1.336

**Table 2 ijms-21-01149-t002:** Values of the coefficients for the estimation of the F-V model with different inputs.

Vel. [m/s]	c(i)	d(i)
~0.0–1.0	−0.4534 *i* ^2^ + 3.7204 *i* + 1.9272	0
~1.0–1.8	0.0773 *i* ^2^ – 0.6573 *i* + 3.3945	−0.4534 *i* ^2^ + 3.7204 *i* + 1.9272
~1.8–2.0	−0.0045 *i* ^2^ + 0.1255 *i* + 3.3809	−0.3916 *i* ^2^ + 3.1946 *i* + 4.6428
~2.0–2.5	−0.0586 *i* ^3^ + 0.5758 i2 – 1.4175 *i* + 1.9272	−0.3925 *i* ^2^ + 3.1675 *i* + 5.319
~2.5–3.25	−0.187 *i* ^2^ + 1.3076 *i* + 0.9671	−0.2724 *i* ^2^ + 2.6697 *i* +6.9998
~3.25–3.5	−0.4325 *i* ^2^ + 1.813 *i* + 2.8962	−0.4127 *i* ^2^ + 3.6504 *i* + 7.7252
~3.5–4.5	0.1386 *i* ^2^ – 1.3686 *i* + 6.9173	−0.5208 *i* ^2^ + 4.1037 *i* + 8.4492
>4.5	2	−0.3822 *i* ^2^ + 2.7351 *i* + 15.366

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
