# Peer review of "Dynamic Analysis of Sphere-Like Iron Particles Based Magnetorheological Damper for Waveform-Generating Test System"

_ijms, 2020, doi:10.3390/ijms21031149_

Round 1

Reviewer 1 Report

While the paper is written well, and presents interesting results, it does not really fir the scope of the journal. The system in hands consists of micron-sized particles, and it is not clear how it is connected to molecular level.

The paper should be transferred to a different journal specializing on vibrations or/and fluid/solid mechanics

Author Response

Answers to the first referee's comments

Article ID: ijms-669245

Title: From micro-sized molecular particles to magnetorheological damper generating the force over 100 kN for shock wave tuning

While the paper is written well, and presents interesting results, it does not really fir the scope of the journal. The system in hands consists of micron-sized particles, and it is not clear how it is connected to molecular level.
The paper should be transferred to a different journal specializing on vibrations or/and fluid/solid mechanics

Ans.: I sincerely thank you for the valuable comments. As you may know, this manuscript was submitted to the special issue entitled “Magnetic-Responsive Molecular Particles Based Smart Materials: Model, Characterization and Applications”. In this special issue, there is a specific topic on the “potential application of MR materials” and “MR materials based sensors and actuators”. You can find it on the website: https://www.mdpi.com/ journal/ijms/special_issues/magnetic_responsive.
In the revised version, in order to accommodate your concern about the scope of the IJMS, we changed the title from “From micro-sized molecular particles to magnetorheological damper generating the force over 100 kN for shock wave tuning” to “Dynamic analysis of sphere-like iron particles based magnetorheological damper for waveform-generating test system” to reflect the intermolecular particle issue. Accordingly, Abstract and Introduction have been modified as follows.

Abstract: In this study, a new double pulse waveform-generating test system with an integrated magnetorheological (MR) damper is proposed. Since the total shear stress of MR fluid can be varied according to shape of particles, sphere-like iron particles based MR fluid is filled into the MR damper. The test system consists of a velocity generator, three masses (impact, test, and dummy), a spring, and an MR damper. To tune the double pulse waveform profile, the damping force model of MR damper is constructed to determine the fundamental parameters of the simulator. Then, the first and second shock waveform profiles are analyzed to solve the governing equation of motions representing the damping force and velocity. The empirical formula of the MR damper is formulated and applied to a simulator with a graphical user interface programmed using MATLAB. The effectiveness of the proposed simulator-featuring controllable MR damper is demonstrated by comparing the simulation and experimental results.
First Paragraph of Introduction: The first discovery of a magnetorheological (MR) fluid was made by Jacob Rainbow in the late 1940s [1-2]. Typically, MR fluids are composed of paramagnetic or soft ferromagnetic particles (~0.03−10 μm) dispersed in a carrier fluid. The MR particles are made of pure iron, carbonyl iron, or cobalt powder and the carrier fluid is a non-magnetic, organic, or aqueous liquid, usually a silicone or mineral oil [3]. Without magnetic field, the MR particles are randomly distributed in the carrier fluid. However, in magnetic field, they undergo rheological phase changes according to the magnitude of the field as shown in Figure 1 (a). The MR particles attain a dipole moment aligned with the external magnetic field and form chains. This chain formation causes an additional yield stress in the fluid, which can be rapidly and continuously controlled by magnetic input. As shown in Figure 1 (b), it is generally known that maximum yield stress of MR fluid is approximately 60-80 kPa. The power requirements for magnetic inputs is up to 50 Watts. In addition, it has been reported that the response time of commercial MR fluid is below 20ms. As a result, MR fluid-based mechanical system have salient advantages such as continuously adjustable dynamic range and fast response. But, the performance of MR fluid can be varied by the particle shape of MR fluid. As shown in Figure 1 (c), the shapes of particles are mostly sphere shape type and plate-lie shape type. Some of the research have implemented to analyze the effect of particle shape of MR fluid [4-7]. From the previous work, it is revealed that sphere-like iron particles based MR fluid is proper for MR damper generating high damping force [8]. So, sphere-like iron particles based MR fluid (RMS Corp., T5-500) is used in this work.

[8] Jiang, Z., & Christenson, R. A comparison of 200 kN magneto-rheological damper models for use in real-time hybrid simulation pretesting. Smart Materials and Structures, 2011, 20(6), 065011.

This point has been well explained in the revised manuscript (page 1, line 2 ~ page 2, line 50).

I sincerely thank you for the valuable comments. I feel that the revised version of the manuscript reflects much better than what was intended in this work.

Reviewer 2 Report

The article « From micro-sized molecular particles to magnetorheological damper generating the force over 100 kN for shock wave tuning » describes the characterization of a new double pulse waveform-generating test system for the evaluation of an integrated magnetorheological (MR) damper.

Despite the tittle of the paper focus more on the damper description, the paper presents the methodology of the realization of a waveform prediction simulator computed using a Matlab application. This means that the tittle should be modified. This is the same for the abstract, what is the real objective of the paper?  To perform a mathematical model of the MR damper or to simulate the acceleration waveform obtained after the shock? Things are not clear.

The introduction part is clear and gives a good understanding of the context of the study. However the authors present the shape of an acceleration pulse waveform after an underwater explosion, what is the link with the MR damper? Figure 2 shows the acceleration shape profile on the test mass but it should be interesting to visualize the impact mass one.

The MR damper experimental results are interesting but it is necessary to complete the study with the damping energy of the damper.

Then the authors perform a modelling of the damping F-V curve using a linearization technique. What is the interest of this paragraph ? It is not clear.

The last part concerns the modelling and the prediction of the « input current …. when there is a desired waveform in a large-scale high-mass impact system. «  the objective is not clear, are the authors trying to develop the command law of the damper for a desired damping force ?

In this paragraph, the comparison between the simulation and experimental results show a real difference and the figure 13 and 14 should be compared using a single figure for each driving current. The comparison between experience and theory should be done with  quantitative indicators.

Finally the conclusion has to be rewritten following all the previous comments.

Minor revisions:

Line 76 : what means EU regulation ?

Line 118 : what properties ?

Line  128 : what is the order of the FIR filter ?

Figure 5 : time and frequency are not visible.

Line 330 : 100kN in the tittle.

Conclusion :

In conclusion the paper needs major revisions and should be rewritten. The authors should choice and describe the objectives of the paper, then the structure of the article will be more clear and provides a much better understanding of the work.

Author Response

Answers to the second referee's comments

Article ID: ijms-669245

Title: From micro-sized molecular particles to magnetorheological damper generating the force over 100 kN for shock wave tuning

The article « From micro-sized molecular particles to magnetorheological damper generating the force over 100 kN for shock wave tuning » describes the characterization of a new double pulse waveform-generating test system for the evaluation of an integrated magnetorheological (MR) damper.
Despite the tittle of the paper focus more on the damper description, the paper presents the methodology of the realization of a waveform prediction simulator computed using a Matlab application. This means that the tittle should be modified. This is the same for the abstract, what is the real objective of the paper? To perform a mathematical model of the MR damper or to simulate the acceleration waveform obtained after the shock? Things are not clear.
: As suggested by the referee, we amended the title, abstract, and introduction as follows:

Title: Dynamic analysis of sphere-like iron particles based magnetorheological damper for waveform-generating test system.
Abstract: In this study, a new double pulse waveform-generating test system with an integrated magnetorheological (MR) damper is proposed. Since the total shear stress of MR fluid can be varied according to shape of particles, sphere-like iron particles based MR fluid is filled into the MR damper. The test system consists of a velocity generator, three masses (impact, test, and dummy), a spring, and an MR damper. To tune the double pulse waveform profile, the damping force model of MR damper is constructed to determine the fundamental parameters of the simulator. Then, the first and second shock waveform profiles are analyzed to solve the governing equation of motions representing the damping force and velocity. The empirical formula of the MR damper is formulated and applied to a simulator with a graphical user interface programmed using MATLAB. The effectiveness of the proposed simulator-featuring controllable MR damper is demonstrated by comparing the simulation and experimental results.
First Paragraph of Introduction: The first discovery of a magnetorheological (MR) fluid was made by Jacob Rainbow in the late 1940s [1-2]. Typically, MR fluids are composed of paramagnetic or soft ferromagnetic particles (~0.03−10 μm) dispersed in a carrier fluid. The MR particles are made of pure iron, carbonyl iron, or cobalt powder and the carrier fluid is a non-magnetic, organic, or aqueous liquid, usually a silicone or mineral oil [3]. Without magnetic field, the MR particles are randomly distributed in the carrier fluid. However, in magnetic field, they undergo rheological phase changes according to the magnitude of the field as shown in Figure 1 (a). The MR particles attain a dipole moment aligned with the external magnetic field and form chains. This chain formation causes an additional yield stress in the fluid, which can be rapidly and continuously controlled by magnetic input. As shown in Figure 1 (b), it is generally known that maximum yield stress of MR fluid is approximately 60-80 kPa. The power requirements for magnetic inputs is up to 50 Watts. In addition, it has been reported that the response time of commercial MR fluid is below 20ms. As a result, MR fluid-based mechanical system have salient advantages such as continuously adjustable dynamic range and fast response. But, the performance of MR fluid can be varied by the particle shape of MR fluid. As shown in Figure 1 (c), the shapes of particles are mostly sphere shape type and plate-lie shape type. Some of the research have implemented to analyze the effect of particle shape of MR fluid [4-7]. From the previous work, it is revealed that sphere-like iron particles based MR fluid is proper for MR damper generating high damping force [8]. So, sphere-like iron particles based MR fluid (RMS Corp., T5-500) is used in this work.

[8] Jiang, Z., & Christenson, R. A comparison of 200 kN magneto-rheological damper models for use in real-time hybrid simulation pretesting. Smart Materials and Structures, 2011, 20(6), 065011.
This point has been well explained in the revised manuscript (page 1, line 2 ~ page 2, line 50).

The introduction part is clear and gives a good understanding of the context of the study. However the authors present the shape of an acceleration pulse waveform after an underwater explosion, what is the link with the MR damper? Figure 2 shows the acceleration shape profile on the test mass but it should be interesting to visualize the impact mass one.
: Since the full-scale submarine or naval vessel testing for heavy-weight components requires many time and cost, BV-43/85 (developed in Germany) rules that a double half-sine acceleration profile which can represent the shock loading caused by an underwater explosion is applied to shipboard or submarine components [28, 29]. As shown in Figure 2, the acceleration profile consists of three parts. In BV-43/85, after first half-sine acceleration waveform (Part I) strikes the test mass, second negative half-sine acceleration waveform (Part II) and very small residual vibration should be applied to test mass (Part III). In particular, the components are mounted on test mass. In order to dissipate the impact energy and realize the required acceleration waveform profile, MR damper with large power dissipation capacity is utilized. After striking, the impact mass is moved in the opposite direction of test mass due to elastic energy of programmer and air spring. Also, when impact mass is decoupled with test mass, the impact mass ideally moves at constant velocity.
This point has been well explained in the revised manuscript (page 3, line 81~line91).
The MR damper experimental results are interesting but it is necessary to complete the study with the damping energy of the damper.
: From Figs. 13 and 14, the magnitude and duration time of 2nd shockwave are increased according to the current input. This is because the reaction force of MR damper is proportional to the damping force of MR damper. Also, the dynamic energy should be dissipated during part III owing to MR damper. In particular, the dynamic energy of testing system is proportional to the acceleration of test mass. From Fig. 14, it can be known that the Part III acceleration results for 0.5 A is reduced from 84. 15m/s2 to 43.87m/s2 in 0.149s.
This point has been well explained in the revised manuscript (page 12, line 329 ~ page 13, line 334).
Then the authors perform a modelling of the damping F-V curve using a linearization technique. What is the interest of this paragraph ? It is not clear.The last part concerns the modelling and the prediction of the « input current …. when there is a desired waveform in a large-scale high-mass impact system. « the objective is not clear, are the authors trying to develop the command law of the damper for a desired damping force ?
: The relationship between magnetic input and shear stress of MR fluid has been researched and established as follows [30, 31]:
                                                         (1)
Where,  and  are the total shear stress and viscosity constant of the MR fluid, respectively.  is the relative shear rate.  is the dynamic yield stress of the MR fluid which are functions of the magnetic field intensity. To generate magnetic field, a DC current input is applied to coil. Accordingly, the total shear stress of MR fluid is affected by the current input and relative shear rate. Especially, some research reported that the viscosity can be tuned by the magnetic field intensity and magnetic field applying time for nanoparticle added MR fluid [32, 33]. Then, the damping force of MR damper is determined by the total shear stress of MR fluid and geometric configuration. However, when big shock is given to MR damper, the response of MR damper is different with that of MR damper with relative low piston velocity. Additionally, the shear rate of MR fluid is dependent on the piston velocity of MR damper. It is judged that the relationship between dynamic yield stress, shear rate and total shear stress is nonlinear. So, to predict the response of MR damper, empirical formula is constructed in this work. The experimental results are divided into intervals with different velocity ranges and linear function modeling based on piston velocity and current input is conducted for each interval. The empirical formula can be utilized as the command law for generating varied double pulse waveform.
[30] Kwon, S. H., Na, S. M., Flatau, A. B., & Choi, H. J. Fe–Ga alloy based magnetorheological fluid and its viscoelastic characteristics. Journal of Industrial and Engineering Chemistry, 2020, 82, 433-438.
[31] Wu, J., Hu, H., Li, Q., Wang, S., & Liang, J. Simulation and experimental investigation of a multi-pole multi-layer magnetorheological brake with superimposed magnetic fields. Mechatronics, 2020, 65, 102314.
[32] Bica, I., Anitas, E. M., Chirigiu, L., Daniela, C., & Chirigiu, L. M. E. Hybrid magnetorheological suspension: effects of magnetic field on the relative dielectric permittivity and viscosity. Colloid and Polymer Science, 2018, 296(8), 1373-1378.
[33] Bica, I., Anitas, E. M., Lu, Q., & Choi, H. J. Effect of magnetic field intensity and γ-Fe2O3 nanoparticle additive on electrical conductivity and viscosity of magnetorheological carbonyl iron suspension-based membranes. Smart Materials and Structures, 2018, 27(9), 095021.
This point has been well explained in the revised manuscript (page 7, line 182 ~ line199).
In this paragraph, the comparison between the simulation and experimental results show a real difference and the figure 13 and 14 should be compared using a single figure for each driving current. The comparison between experience and theory should be done with quantitative indicators.
: The measured magnitude of 1st shockwave is some different from that of predicted shockwave. During impact between impact and test mass, the high-impact velocity is unequally applied to test mass. That is reason of the difference at 1st shockwave. The maximum error value of 1st shockwave is 173.14m/s2 at 0A. The other tangible difference is the duration time of 2nd shockwave (t2). The maximum difference value of 2nd shockwave is 0.01s at 0A. This is related with nonlinearity of elastic and damping force for sudden impact and slow response time of MR damper.
This point has been well explained in the revised manuscript (page 13, line 337 ~ line 340).

Finally the conclusion has to be rewritten following all the previous comments.
: As suggested by the referee, we amended the conclusion as follows:
In this study, we proposed an MR damper modeling and a waveform prediction program to simulate the double pulse from an underwater explosion. Impact experiments were carried out to obtain the damping F-V curve, and that with the highest velocity was modeled mathematically. For this, the curve with high nonlinearity was divided into intervals with different velocity ranges, with rapid variations in the damping force. A linear function modeling was conducted for each interval. Then, the simulation results were compared with the actual impact experiment results. From the comparison results between simulation and experimental results, the proposed dynamic modeling of system and damping are quite self-explanatory, showing that the good agreement with experimental results. In addition, it has been confirmed that MR damper featuring sphere-like iron particles shows high damping force up to 200 kN. Since the performance on damping force, durability, sedimentation and dispersible stability are depending upon on the particle shape, MR dampers consisting of different particle shape need to be more investigated to find optimal performance for the specific objective in the application. This investigation will be explored as a second phase in the future.

This point has been well explained in the revised manuscript (page 15, line 363 ~ line 367).

Minor revisions:
Line 76 : what means EU regulation ?
: For a generic underwater situation, the shock testing is defined by an EU regulation, which defines the most basic waveform as a double pulse that can be expressed using acceleration [27, 28].--> Since the full-scale submarine or naval vessel testing for heavy-weight components requires many time and cost, BV-43/85 (developed in Germany) rules that a double half-sine acceleration profile which can represent the shock loading caused by an underwater explosion is applied to shipboard or submarine components [28, 29].
This point has been well explained in the revised manuscript (page 2, line 81~line91).

Line 118 : what properties ?
Ans.: Depending on the objective of the experiment, the variable properties were determined by varying the current applied to the MR damper while fixing the impact mass, impact velocity, shock programmer stiffness, and test table mass. à Depending on the objective of the experiment, the damping force tuning properties of MR damper were evaluated by varying the current applied to the MR damper while fixing the impact mass, impact velocity, shock programmer stiffness, and test table mass.)
This point has been well explained in the revised manuscript (page 5, line 130 ~ line 133).

Line 128 : what is the order of the FIR filter ?
Ans.: The order of FIR filer is 100.

This point has been well explained in the revised manuscript (page 5, line 141 ~ 142).

Figure 5 : time and frequency are not visible.
Ans.: As suggested by the referee, we amended the Figure 5 (page 5, line 157)
Line 330 : 100kN in the tittle.
Ans.: As suggested by the referee, we amended the title of this manuscript.

Conclusion :
In conclusion the paper needs major revisions and should be rewritten. The authors should choice and describe the objectives of the paper, then the structure of the article will be more clear and provides a much better understanding of the work.

Ans.: As suggested by the referee, we amended the manuscript. The modification is highlighted in red color.

I sincerely thank you for the valuable comments. I feel that the revised version of the manuscript reflects much better than what was intended in this work.

Reviewer 3 Report

It is well known that the viscosity plays an important role when an external magnetic field is applied, see for example: https://doi.org/10.1007/s00396-018-4356-1 or https://doi.org/10.1088/1361-665X/aad5a6 . How this  affects the performance of the MR damper?  Please clarify the meanings of the figures labels. For example, what means “0A”, “1A”, “2A”, “3A”? In Figs. 6, 7 etc. ? They are the values of the input current ( ) through the coil of the damper? If yes, then what is the dependence ? In “Conclusion” section you mention “Since the yield stress of MR fluid can be varied according to shape of particles, the performance of MR dampers with several particle shape will  be evaluated as a second future work”. However, you include Fig. 1c, which shows the shape of the particles. It will be useful, if possible, to give some explanations concerning the impact of the microparticle shapes on the response functions of the dampers In Figure 5a and b, the labels “Time” and “Frequency” are not clearly seen. Please correct. Insert a vertical space between the lines 208 and 209. Throughout the manuscript, the equations could be written more compact, but this is up to the authors. The list of References shall include more recent works which reflect, in a more balanced way, relevant results of other groups.

Author Response

Answers to the third referee's comments

Article ID: ijms-669245

Title: From micro-sized molecular particles to magnetorheological damper generating the force over 100 kN for shock wave tuning

It is well known that the viscosity plays an important role when an external magnetic field is applied, see for example: https://doi.org/10.1007/s00396-018-4356-1 or https://doi.org/10.1088/1361-665X/aad5a6. How this affects the performance of the MR damper?
: The magnetorheological fluid is the suspension of iron particles dispersed in a non-magnetic medium. The properties of the MR fluid without magnetic field are, but the unrestrained particles transform to a chain-like structuring under an external magnetic field. A relationship between magnetic input and shear stress of MR fluid has been researched and established as follows [30, 31]:
                                                          (1)
where  and  are the total shear stress and viscosity constant of the MR fluid, respectively.  is the relative shear rate.  is the dynamic yield stress of the MR fluid which are functions of the magnetic field intensity. To generate magnetic field, a DC current input is applied to coil. Accordingly, the total shear stress of MR fluid is affected by the current input and relative shear rate. Especially, some research reported that the viscosity can be tuned by the magnetic field intensity and magnetic field applying time for nanoparticle added MR fluid [32, 33]. Then, the damping force of MR damper is determined by the total shear stress of MR fluid and geometric configuration. However, when big shock is given to MR damper, the response of MR damper is different with that of MR damper with relative low piston velocity. Additionally, the shear rate of MR fluid is dependent on the piston velocity of MR damper. It is judged that the relationship between dynamic yield stress, shear rate and total shear stress is nonlinear. So, to predict the response of MR damper, empirical formula is constructed in this work. The experimental results are divided into intervals with different velocity ranges and linear function modeling based on piston velocity and current input is conducted for each interval. The empirical formula can be utilized as the command law for generating varied double pulse waveform.
[30] Kwon, S. H., Na, S. M., Flatau, A. B., & Choi, H. J. Fe–Ga alloy based magnetorheological fluid and its viscoelastic characteristics. Journal of Industrial and Engineering Chemistry, 2020, 82, 433-438.
[31] Wu, J., Hu, H., Li, Q., Wang, S., & Liang, J. Simulation and experimental investigation of a multi-pole multi-layer magnetorheological brake with superimposed magnetic fields. Mechatronics, 2020, 65, 102314.
[32] Bica, I., Anitas, E. M., Chirigiu, L., Daniela, C., & Chirigiu, L. M. E. Hybrid magnetorheological suspension: effects of magnetic field on the relative dielectric permittivity and viscosity. Colloid and Polymer Science, 2018, 296(8), 1373-1378.
[33] Bica, I., Anitas, E. M., Lu, Q., & Choi, H. J. Effect of magnetic field intensity and γ-Fe2O3 nanoparticle additive on electrical conductivity and viscosity of magnetorheological carbonyl iron suspension-based membranes. Smart Materials and Structures, 2018, 27(9), 095021.

This point has been well explained in the revised manuscript (page 7, line 182 ~ line199).

Please clarify the meanings of the figures labels. For example, what means “0A”, “1A”, “2A”, “3A”? In Figs. 6, 7 etc. ? They are the values of the input current ( ) through the coil of the damper? If yes, then what is the dependence ?
: Yes, in order to generate magnetic field, a DC current is applied to coil. This point has been well explained in the revised manuscript (page 7, line 189). Also, the relation between input current and MR fluid has been already answered in Q. 1.
In “Conclusion” section you mention “Since the yield stress of MR fluid can be varied according to shape of particles, the performance of MR dampers with several particle shape will be evaluated as a second future work”. However, you include Fig. 1c, which shows the shape of the particles. It will be useful, if possible, to give some explanations concerning the impact of the microparticle shapes on the response functions of the dampers.
: The magnetorheological fluid is the suspension of iron particles dispersed in a non-magnetic medium. The properties of the MR fluid without magnetic field are, but the unrestrained particles transform to a chain-like structuring under an external magnetic field. A relationship between magnetic field and shear stress of MR fluid has been researched and established as follows [??]:
(1)
where  and  are the total shear stress and viscosity constant of the MR fluid, respectively.  is the relative shear rate.  is the dynamic yield stress of the MR fluid which are functions of the magnetic field intensity. And are different according to shape of iron particle.
This point has been well explained in the revised manuscript (page 7, line 182 ~ line199).
In Figure 5a and b, the labels “Time” and “Frequency” are not clearly seen. Please correct.
: As suggested by the referee, we amended the Figure 5 as follows:

(b)

Figure 5. Measured damping force in the time and frequency domains: (a) Measured damping force in the time domain;(b) Measured damping force in the frequency domain.

Insert a vertical space between the lines 208 and 209.
: As suggested by the referee, we amended the manuscript.
Throughout the manuscript, the equations could be written more compact, but this is up to the authors.
: As suggested by the referee, we amended the manuscript.
The list of References shall include more recent works which reflect, in a more balanced way, relevant results of other groups.

Ans.: As suggested by the referee, we added recent references as follows:
[8] Jiang, Z., & Christenson, R. A comparison of 200 kN magneto-rheological damper models for use in real-time hybrid simulation pretesting. Smart Materials and Structures, 2011, 20(6), 065011.

[30] Kwon, S. H., Na, S. M., Flatau, A. B., & Choi, H. J. Fe–Ga alloy based magnetorheological fluid and its viscoelastic characteristics. Journal of Industrial and Engineering Chemistry, 2020, 82, 433-438.
[31] Wu, J., Hu, H., Li, Q., Wang, S., & Liang, J. Simulation and experimental investigation of a multi-pole multi-layer magnetorheological brake with superimposed magnetic fields. Mechatronics, 2020, 65, 102314.
[32] Bica, I., Anitas, E. M., Chirigiu, L., Daniela, C., & Chirigiu, L. M. E. Hybrid magnetorheological suspension: effects of magnetic field on the relative dielectric permittivity and viscosity. Colloid and Polymer Science, 2018, 296(8), 1373-1378.
[33] Bica, I., Anitas, E. M., Lu, Q., & Choi, H. J. Effect of magnetic field intensity and γ-Fe2O3 nanoparticle additive on electrical conductivity and viscosity of magnetorheological carbonyl iron suspension-based membranes. Smart Materials and Structures, 2018, 27(9), 095021.

I sincerely thank you for the valuable comments. I feel that the revised version of the manuscript reflects much better than what was intended in this work.

Round 2

Reviewer 2 Report

The answers to my comments are satisfactory. The paper can be accepted in the actual form.